# Exploring the complex nature of implementation of Artificial intelligence in clinical practice: an interview study with healthcare professionals, researchers and Policy and Governance Experts

Jobbe P.L. Leenen[1,2]*, Paul Hiemstra[2], Martine M. Ten Hoeve[2], Anouk C.J. Jansen[2], Joris D. van Dijk[3], Brian Vendel[3], Guido Versteeg[4], Gido A. Hakvoort[2], Marike Hettinga[2]

**1** Connected Care Center, Isala, Zwolle, Overijssel, The Netherlands, **2** Research Group IT Innovations in Healthcare, Windesheim University of Applied Sciences, Zwolle, Overijssel, The Netherlands, **3** Department of Nuclear Medicine, Isala, Zwolle, Overijssel, The Netherlands, **4** Appbakkers, Zwolle, Overijssel, The Netherlands

* j.p.l.leenen@isala.nl

## Abstract

Artificial Intelligence (AI)-based tools have shown potential to optimize clinical workflows, enhance patient quality and safety, and facilitate personalized treatment. However, transitioning viable AI solutions to clinical implementation remains limited. To understand the challenges of bringing AI into clinical practice, we explored the experiences of healthcare professionals, researchers, and Policy and Governance Experts in hospitals. We conducted a qualitative study with thirteen semi-structured interviews (mean duration 52.1 ± 5.4 minutes) with healthcare professionals, researchers, and Policy and Governance Experts, with prior experience on AI development in hospitals. The interview guide was based on value, application, technology, governance, and ethics from the Innovation Funnel for Valuable AI in Healthcare, and the discussions were analyzed through thematic analysis. Six themes emerged: (1) demand-pull vs. tech-push: AI development focusing on innovative technologies may face limited success in large-scale clinical implementation. (2) Focus on generating knowledge, not solutions: Current AI initiatives often generate knowledge without a clear path for implementing AI models once proof-of-concept is achieved. (3) Lack of multidisciplinary collaboration: Successful AI initiatives require diverse stakeholder involvement, often hindered by late involvement and challenging communication. (4) Lack of appropriate skills: Stakeholders, including IT departments and healthcare professionals, often lack the required skills and knowledge for effective AI integration in clinical workflows. (5) The role of the hospital: Hospitals need a clear vision for integrating AI, including meeting preconditions in infrastructure and expertise. (6) Evolving laws and regulations: New regulations can hinder AI development due to unclear implications but also enforce standardization, emphasizing quality and safety in healthcare. In conclusion, this study highlights the complexity of AI implementation

**Data availability statement:** Due to privacy concerns related to the small sample size, the specificity of the study context, and the personal data involved, data transcripts can not be openly available. Public deposition would breach compliance with the protocol approved by our research ethics board. However, data can be accessed upon reasonable request at the independent data manager Iris Goes of Windesheim University of Applied Sciences by email: i.goes@windesheim.nl.

**Funding:** This study was financially supported by: TechForFuture (TFF) "Centre of Expertise of Applied High Tech Systems and Materials (HTSM) Oost", a cooperation of the province of Overijssel and the University of Applied Sciences Windesheim and Saxion, project "MEDIATE: a medical data analytics center", Project Number TFF2217. Research group IT innovations in Healthcare of University of Applied Sciences Windesheim received the grant. The funders had no role in study design, data collection and analysis, decision to publish, or preparation of the manuscript.

**Competing interests:** The authors have declared that no competing interests exist.

in clinical settings. Multidisciplinary collaboration is essential and requires facilitation. Balancing divergent perspectives is crucial for successful AI implementation. Hospitals need to assess their readiness for AI, develop clear strategies, standardize development processes, and foster better collaboration among stakeholders.

## Author summary

We explored the challenges of implementation of Artificial Intelligence (AI) into clinical practice in hospitals by interviewing healthcare professionals, researchers, and Policy and Governance Experts. Despite AI's potential to improve patient care and streamline clinical processes, its implementation in hospitals is still limited. Through thirteen in-depth interviews, we identified several key issues. First, we found that AI initiatives often focus more on creating innovative technology rather than addressing actual clinical needs, leading to limited success in real-world application. Additionally, many initiatives aim to generate knowledge without clear plans for practical implementation. Effective collaboration is also lacking, as diverse stakeholders are either not involved or involved too late, making communication difficult. We noticed a significant gap in necessary skills and knowledge among both IT staff and healthcare providers, which hinders AI integration. Hospitals also need a clear vision and adequate infrastructure to support AI. Finally, evolving laws and regulations present both challenges and opportunities, as they can complicate development but also drive the standardization and safety of AI tools. In conclusion, implementing AI in clinical settings is complex and multifaceted. Hospitals need to assess their readiness for AI, develop clear strategies, standardize development processes, and foster better collaboration among stakeholders.

## Introduction

Artificial intelligence (AI) has emerged as a promising tool to address the growing challenges in healthcare, such as an aging population and staff shortages [1,2]. AI-based tools have demonstrated the potential to optimize clinical workflows, enhance patient safety, aid in diagnosis, and facilitate personalized treatment [3]. All of this with the potential of lowering the amount work-hours required, which is important given the current capacity shortages in healthcare [4]. Several cutting-edge AI technologies have shown the potential to transform hospital care in several fields. One of the leading fields is diagnostics by medical imaging, for instance, in detection and prediction of cancer, COVID-19 identification, diabetes and cardiology in countries across Asia and Europe [5–10]. More than half of CE-marked (Conformité Européenne) medical AI devices between 2015 and 2020, certified for compliance with EU safety and performance standards, were intended for use in imaging [11]. In addition, progress is being made in the areas of

optimalisation of surgical candidacy, management of therapeutic options and rehabilitation by the use of AI models and is increasing rapidly [12,13].

Despite the promises of AI for patients and the increase of published articles, its successful integration into clinical practice remains limited [14–18]. These discrepancies between the high potential of AI and its actual implementation in healthcare raises important questions about the barriers to its implementation. One major reason for the gap between the promise of AI and its clinical implementation seems the lack of a broad, holistic view in the development of AI models [19]. Many AI development initiatives focus narrowly on developing an AI model, sometimes with poorly defined use cases [18], and without considering the complex healthcare environment in which these technologies will ultimately operate. This includes the complexity of the IT infrastructure,(16) the lack of automated data routing and processing [17], and the limited infrastructure to deploy and scale these AI applications [14]. This results in solutions that are not aligned with the needs and workflows of healthcare professionals and other staff, hindering their acceptance and adoption [20].

To address this gap between promise of AI and its clinical implementation, a framework has been proposed to guide AI initiatives from ideation to clinical implementation. This framework, the Innovation Funnel for Valuable AI in Healthcare [21] (IFVAIH) (Dutch: Waardevolle AI tool) (Fig 1), offers a comprehensive checklist for researchers and developers in advancing valuable applications from inception to scalability, this tool provides guidance to the legal and regulatory parameters governing their actions. This allows an early start to prepare for minimum requirements or standards and to reflect on actions to proceed in the IFVAIH. In the seven-phase funnel, which includes the five pervasive domains of Value (ensuring AI initiatives provide clear benefits), Application (practical use of AI in clinical settings), Ethics (ethical considerations and adherence to ethical standards), Technology (leveraging technological capabilities), and Responsibility (accountability and responsibility of AI use), each phase has a clear goal and encourages creative work while using resources wisely,

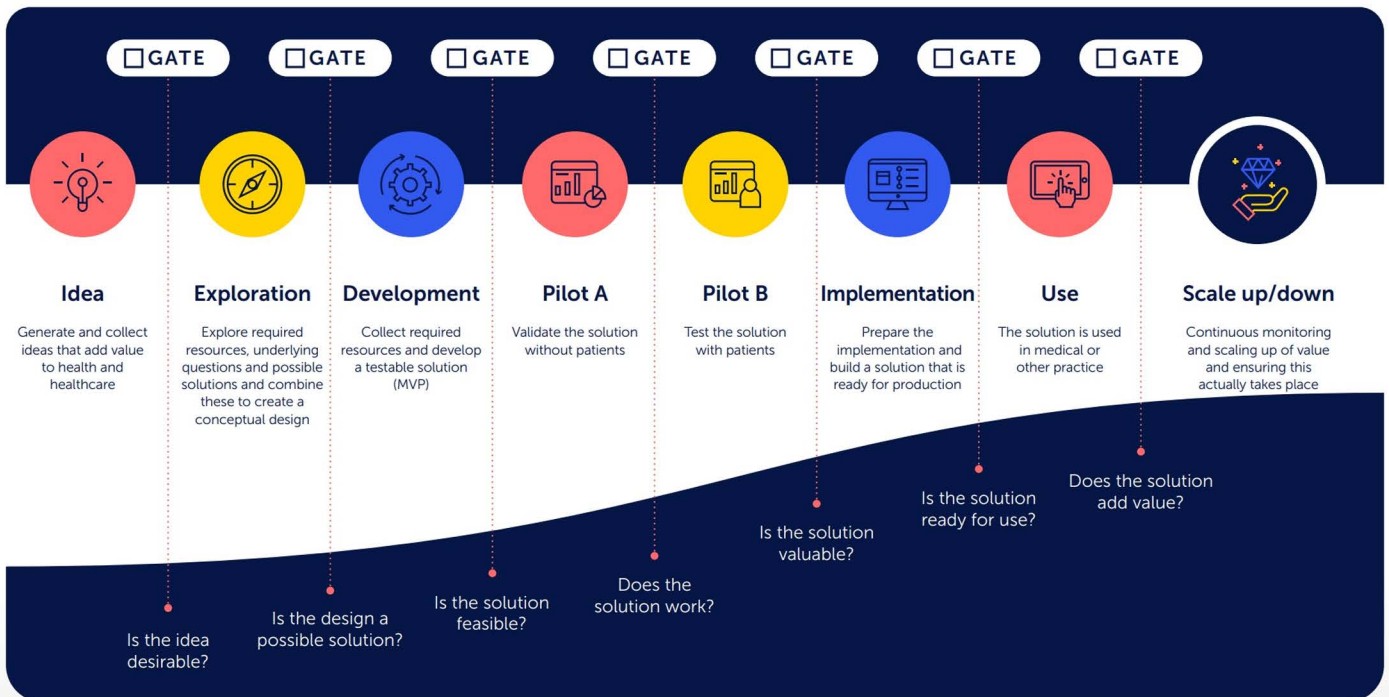

**Fig 1. Phases of the Innovation Funnel for Valuable AI in Healthcare [21] (IFVAIH).** Abbreviations: AIPA: Artificial Intelligence Prediction Algorithm.

providing a structured approach that helps strategize to meet standards and develop human-centric AI applications for clinical practice.

While the IFVAIH framework provides a technical approach to integrating valuable AI solutions in hospitals, it remains unclear what practical implementation methods are needed beyond purely technical aspects to achieve successful clinical implementation. As a first step in developing a broad, holistic view, it is essential to understand the experiences and perspectives of stakeholders in the implementation of AI in hospitals in order to identify challenges and opportunities in this process. Hence, the aim of the study was to explore the experiences of healthcare professionals, researchers, and Policy and Governance Experts in AI implementation in hospitals. Specifically, we aim to identify the challenges encountered by these stakeholders in developing and implementing AI solutions.

## Materials and methods

### Design

We performed a qualitative study design by semi structured interviews. This study is reported in concordance with the consolidated criteria for reporting qualitative research (COREQ) [22].

### Setting and participants

The study was conducted in a 942-bed tertiary teaching hospital in the Netherlands with ample experience in AI development. Participants with experience with AI development, research and/or implementation in their practice were eligible to be interviewed. We defined three subgroups of participants: healthcare professionals (physicians, nurses), researchers (medical technology, technical medicine, data scientists, PhD students) and Policy and Governance Experts (members of the board, managers, policy advisors, legal experts). Inclusion criteria were having experience with at least one AI initiative as a use case of AI development in their current job position/specialism, and willingness to participate in the study. To explore a broad range of experiences with AI, we interviewed a purposefully sampled group of employees per subgroup.

### Participant selection

Participants were purposeful sampled through the network of the researchers and contacted by email; a reminder was sent if there was no response within two weeks. The study was not advertised by regular communication channels of the hospital. Although participants worked in the same hospital (of approximately 7000 employees), there were no direct or prior colleagues or relationships with the participants. A minimum of four participants per subgroup was deemed sufficient to explore the major experiences and reach data saturation. We scheduled the interviews after informed consent.

### Data collection

Semi-structured, online interviews (Microsoft Teams) were conducted with the participants between November 2023 and January 2024 by JL and MtH. Both had prior experience in qualitative research.Before forming the interview guide and formal data collection, we conducted three open explorative interviews with two AI developers and one AI researcher. Based on these interviews and the IFVAIH(21) (Fig 1) we developed the interview guide consisting of 19 questions over five domains of the IFVAIH Funnel: value, application, ethics, technology, responsibility (S1 Appendix). The figure of the AI funnel was used as visual aid in the introduction of the interview. The interview guide was pilot tested with one of the AI developers.

While the interviewers adhered to the structure provided by the guide, they were also permitted to adjust the sequence of questions within each topic and incorporate supplementary questions to address emerging themes. Notably, interviewers were guided by the interview guide, but were allowed to change the sequence of questions within the topics or allow additional questions for emerging topics. Different probing techniques such as remaining silent, echoing, and asking

for elaboration were used to gain further insight into specific experiences. All interviews were audiovisual-recorded and transcribed verbatim using the built-in tools in Microsoft Teams. Keynotes were used to record feelings and thoughts of the researchers.

### Data analysis

To analyse the interviews, we used a six-step thematic analysis [23] using the qualitative data analysis software ATLAS. ti 23.0 (Cleverbridge AG, Cologne, Germany). We used this approach to maintain openness and flexibility in identifying emerging themes based on the codes during the analysis of the data collected. The stages included: (1) immersion; (2) generating initial codes; (3) searching for and identifying themes; (4) reviewing themes; (5) defining and naming themes; and (6) writing the report. Stage 1 to 3 were conducted independently by two researchers (JL and MtH). During the first and second stage, JL (clinical background) and MtH (technical background) became familiar with the data by listening to the audiovisual recordings, checking the transcriptions against the audio recording, reading, listening sections again and re-reading the final transcripts. During the third stage, both researchers read the transcripts and codes for categorizing similar statements into first themes. For the fourth and fifth stages, JL, MtH, PH (technical background), AJ (clinical background), GH (technical background) were responsible for reviewing, defining and naming themes, which were discussed with all authors. Also the formed themes were mapped onto the five domains (value, application, ethics, technology, and responsibility) of the IFVAIH funnel. The diverse background of the interview moderators and researchers ensured a comprehensive and holistic perspective on AI implementation. The clinical background of the researchers likely facilitated rapport and recognition among participants, thereby fostering open discussions about clinical challenges and opportunities. Conversely, the technical background likely encouraged in-depth conversations on technology and data-related topics. During the sixth stage, the themes were brought to participants for member checking by e-mail, which did not result in any changes to the themes.

## Results

In total, we approached 14 professionals of which 13 responded and were interviewed with a median duration of 51.7 minutes (interquartile 49.3-55.8). Two Policy and Governance Experts (IDs 9 and 10) were also working part-time as healthcare professionals (Table 1). From these interviews we extracted six key themes: 1) Balancing between demand-pull and tech-push, 2) Focus on generating knowledge, not implementing working solutions, 3)Lacking multidisciplinary collaboration, 4) Lack of appropriate knowledge and skills, 5) The role of the hospital in AI development and 6) Opportunities and challenges from evolving laws and regulations (Table 2). All domains of the IFVAIH Funnel were found in the themes.

### Theme 1: Balancing between demand-pull and tech-push

In the first theme, participants mentioned the challenge of not consistently starting from clinically relevant demands but starting from an inspiring example of an AI technology. The push to experiment with the AI technology came from a technological perspective: can this specific task be done using AI? By 'demand-pull,' we refer to innovation driven by user needs, while 'tech-push' refers to innovation driven by technological advancements without explicit demand from end-users. The following quote illustrates the 'tech-push' approach, where the drive to use AI originated from ongoing technological efforts rather than a clinical demand: *'We were already working on AI and then we thought that it might also have potential for better diagnosis in the other patient group. (…) However, the potential room for improvement of our current method for this other patient group seem to be minimal '*(Participant #2, healthcare professional).

They also indicated that commercial parties had this tendency to tech-push. The companies look at the use of AI for diagnosis of diseases from a more commercial point of view. According to them, patient volumes are a determining factor for the viability of AI tools, which can clash with challenges experienced by healthcare professionals. A participant

**Table 1.  Participants' characteristics.**

| Subgroup | ID# | Job position | Specialism | Sex |
|---|---|---|---|---|
| Healthcare professionals | 1 | Physician | Nuclear medicine | Male |
| | 2 | Physician | Radiology | Male |
| | 3 | Physician | Gastroenterology | Male |
| | 4 | Physician | Radiotherapy/oncology | Male |
| Researchers | 5 | Technical physician/PhD student | Radiology | Male |
| | 6 | Technical physician/PhD student | Radiology | Male |
| | 7 | Data scientist | Multiple disciplines | Male |
| | 8 | Data scientist | Multiple disciplines | Male |
| Policy and Governance Experts | 9 | Chief Medical Information Officer, Clinical Physicist | Medical technology | Female |
| | 10 | Nurse Information Officer, Medium Care Nurse | Intensive Care | Male |
| | 11 | Member of the board | n/a | Male |
| | 12 | Medical ethicist | n/a | Male |
| | 13 | Legal expert | n/a | Female |

Abbreviations: ID#: identification number, n/a: not applicable.

**Table 2.  Overview of themes, subcategories and mapping onto the IFVAIH Funnel.**

| Theme | Subcategories | Domains of the IFVAIH |
|---|---|---|
| Balancing between demand-pull and tech-push | Tech-push by examples and commercial parties | Value |
| | Experience with AI needed for demand-pull | Value |
| | Demand-pull factors | Value |
| Focus on generating knowledge, not implementing working solutions | Unclear goal definition and required steps for implementation | Value, Application |
| | Focus on research vs. focus on clinical practice | Value, Application |
| Lacking multidisciplinary collaboration | Stakeholders are not or only involved in later stages | Application, Technology, Ethics, Responsibility |
| | Variety of perspectives on AI development and implementation | Application, Technology, Ethics, Responsibility |
| Lack of appropriate knowledge and skills | Of requirements of AI development | Technology |
| | Of ethical, legal and societal aspects of AI development | Ethics, Responsibility |
| | Of added value in clinical practice of AI development | Value, Application |
| The role of the hospital in AI-development | Vision and strategic planning | Value, Reponsibility |
| | Technical requirements and necessary infrastructure | Technology |
| | Responsibility as AI manufacturer | Responsibility |
| Opportunities and challenges from evolving laws and regulations | Increasing safety and reliability | Responsibility, Ethics |
| | Lack of clarity and unfamiliarity with new regulations | Responsibility |
| | Integration in clinical guidelines | Application, Responsibility |

Abbreviations: IFVAIH: Innovation Funnel for Valuable AI in Healthcare.

said: '*Yes, commerce is just profit-driven, so they just want to maximise profits. But we want to know how it works and how we can best help the patient. And yes, that's where the difference is.*' (Participant #5, researcher).

In addition to a tendency towards tech-push from both commercial vendors and AI researchers, the participants felt that it was hard for them to accurately voice their needs in a manner that is useful for AI developers. This limits the demand-pull they can exert on companies and AI researchers. Participants indicated that this was caused by their unfamiliarity with AI, linked to it in the early stages of showing its potential for health care. Fortunately, they feel that as soon as more

healthcare professionals realize what AI can mean for their current practice, they will be more inclined to invest time in understanding AI, improving their ability to voice their demands accurately. One healthcare professional said about this: *'It just takes time to gain experience with it. And once you have that experience, you can see much more clearly see the potential of the tool to deliver better care. You didn't see that before because you just didn't know how it worked.'* (Participant #3, healthcare professional)

Furthermore, researchers and healthcare professionals mentioned the tech-push may lead to the risk of non-valuable care delivery by the implementation of too inaccurate AI models. One participant said: *'This tech-driven approach can be problematic. A lot of physiological processes can be normal and not necessarily a sign of disease, right? For example, polyps: 90% of polyps are benign and certain benign polyps need treatment while the majority don't. So, yes, if the AI tool for detection of maligne polyps is not accurate enough, it will lead to overdiagnosis rather than underdiagnosis and eventually overtreatment.'* (Participant #3, healthcare professional)

Another impact of the focus on tech-push is the lack of integration of the technology into the existing clinical workflow. This lack of integration severely limits the potential adoption of AI technologies in clinical practice. One participant said: *'if you want clinical adoption, you just have to make it felt. This means that the workflow must be seamless and here we come to the big point. Most AI products have a poor workflow.'* (Participant #2, healthcare professional)

In addition, a demand-pull factor for healthcare professionals to get involved in AI development could be when the declining quality of care and longer waiting lists due to staff shortages and increase of patient volumes could add to the urgency of incorporating AI to maintain the accessibility of care for their patients. One participant said about demand-pull for patients and professionals*: 'Staff shortage plays a big role here. What would you do when there is no staff to help you with your complaint? Receiving no diagnosis at all or an AI diagnosis?' I think then majority of patients and professionals would be preferring an AI diagnosis.* (Participant #9, Policy and Governance Expert). Another healthcare professional said the following about demand-pull from professional perspective: *'Take image recognition as an example. I have had to wait until 12 a.m. for the radiologist's results. If we could rely on an AI assessment, I could have gone home much sooner and I experience less workload.'* (Participant #3, healthcare professional)

In relation to the previous quote, another demand-pull factor for use of AI would be the backup offered to healthcare professionals by an AI system. They see AI as a backup system of their own clinical decision. Certainly this would be relevant during certain moments where the risk of error is increased. One participant said: '*It offers me more ease of mind if I am called in during the shift in the middle of the night by the physician assistant to assess a radiological image. As you can imagine, you are less alert than during the day and the chances of making a wrong assessment are much higher. Having the AI watching along with my assessment does give me peace of mind.*' (Participant #2, healthcare professional)

## Theme 2: Focus on generating knowledge, not implementing working solutions

When AI research initiatives start in a grassroots manner, often the exact clinical goal is not clearly defined. This clinical goal is not just a general statement of the added value, but a detailed breakdown of how it will impact clinical practice, and a plan on how to build towards implementation. One participant said: '*Yes you start something with a good idea, and it keeps growing and then suddenly has a lot of potential in patient care. You just haven't thought through beforehand how you get from a nicely written article to the next step and who you need in the process.*' (Participant #6, researcher). This explains why the majority of proof-of-concept's do not grow into implemented AI solutions.

When goals are more defined, two distinct goals were mentioned: developing AI models for research use only or for implementation in practice. Most participants developed models for research use, in essence delivering a proof-of-concept whether or not a particular task could be performed by an AI. The researchers and healthcare professionals point out that developing and validating a well-functioning AI model is quite a task in itself, for example in terms of obtaining a sufficient number of samples and the time required to label the data. They point out that these steps are essential to take a model to the next stage of implementation. One participant said: *'Before you have a good algorithm, you have to validate it all the*

*time with all these images. Yes, that takes a lot of hours from experienced doctors. All the images are already blinded and coded, and if I have to check, say, 4,000 images, yes, it takes me at least 3-4 minutes per image. (...) You have to get this step right if you want to move on to implementation.'* (Participant #3, healthcare professional)

The step from a proof-of-concept AI model to practical implementation was not obvious to clinicians and researchers. They are unaware of the various steps and requirements or are not clearly defined, especially in the area of technical requirements and integration with existing IT infrastructure, as well as legal and privacy considerations.

### Theme 3: Lacking multidisciplinary collaboration

Most of the participants indicated their AI initiatives to lack multidisciplinary collaboration with other experts. Several participants admitted to not involve relevant stakeholders in their initiatives. As one participant said: *'it is not clear what key issues you need to think about if you want to come from having developed a good model, to using it in practice. (...) No I have not involved anyone from the IT department or a legal expert.'* (Participant #4, healthcare professional). This raises the issue of understanding why such collaboration is often overlooked and underscores the need to identify the underlying causes of this gap in teamwork and communication across disciplines

The medical AI developments often show the pattern of a 'hobby project', involving a doctor and a technical partner. While they often show genuine enthusiasm for AI technology, they lack a partner whose primary focus is on the real-world clinical impact of their research. This lack of a multidisciplinary approach was well reflected in the following quote: *'It arises together with a great idea to use AI to improve healthcare. The focus is then really on developing a good model that has potential. (...) We did not yet involve technicians in the project, that was something for later we thought.'* (Participant #9, Policy and Governance Expert)

When multidisciplinary collaboration does take place, essential partners are involved too late in the process. These partners include legal experts, IT departments, patients, and workplace end-users. For example, one participant said: *'Yes, we were well into development and ready for the next step, but it turns out that we need a lot of resources from the technical department to run the model in the hospital infrastructure. There are costs associated with that that we had not thought about before.'* (Participant #5, researcher)

Once stakeholders are engaged in collaboration, challenges arise due to their different perspectives on the AI initiatives. For example, while AI researchers have an intrinsic fascination with AI and strive for scientific advancement and publication, doctors and nurses are mainly focused on the immediate added value on their work and patients. In contrast, commercial parties are driven by concerns for scalability and profitability, while legal departments are primarily concerned with compliance to legal frameworks. These differing perspectives are well reflected in the following quotes: *'I think nurses are mostly concerned, if they even are concerned about the potential of AI at all, with what it brings to their day-to-day work'* (Participant #10) and *'There are just a lot of divergent interests and that can sometimes be complicating bringing them together to create real impact in the field.'* (Participant #12, Policy and Governance Expert).

Collaboration with commercial partners has its own set of challenges. One participant said: *'Doing the research properly takes time and therefore money. However, for the commercial parties, it is important that the costs are also going to be covered by income based on the developed models during the studies. As a result, there are differences in the desired and actual speed of AI projects.'* (Participant #9, Policy and Governance Experts) Moreover, according to the participants, this may also result in implementable AI models being developed only for high-volume patient groups and the model being scoped very specifically for one group, while the improvement potential for this group in terms of quality of care is less significant and clinically relevant. This was reflected in the following quote: *'it keeps you focused on point solutions whereas if you really want to make an impact, you have to think in terms of the breadth of the entire patient care process. Even in the parts where less money can potentially be made.'* (Participant #12, Policy and Governance Experts)

## Theme 4: Lacking the appropriate knowledge and skills

An important aspect affecting multidisciplinary collaboration are a lack of required knowledge and skills among stakeholders. First, IT departments may struggle with the requirements to implement AI solutions at the operational level as IT resources are not organized to integrate models in clinical workflows. One participant said: *'I also think the IT department does not have the sufficient knowledge and experience with AI in their hospital. As a result every AI project has to incur costs whereas this may be a centralized budget.'* (Participant #2, healthcare professional). Second, AI researchers wonder how to conduct research that is both academically rigorous and compliant with Medical Device Regulation (MDR) legislation. This is well reflected in the quote of a researcher*: 'the laws and regulations for research are clearer for us as researchers. The legislation of the MDR is much less known to us whereas you might actually want to be compliant with both legislations from the start.'* (Participant 7, researcer). Third, legal teams face the challenge of translating the complex regulations surrounding AI-act and MDR into practical guidelines for hospitals, including the definitions and regulations related to in-house development of medical products. The participant said*: 'Legislation may have been drafted, but how it is then translated into daily practice, that is then for the hospitals to decide.'* (Participant #13, Policy and Governance Expert). Fourth, nursing staff still are not familiar with the potential of AI for the nursing care, despite AI solutions potentially having a broader multidisciplinary impact beyond the medical field. This lack of involvement not only hampers the development of data-driven practices but also delays adoption of AI within nursing care. One nurse said: *'I think AI lends itself very well to nursing issues. Especially also solutions at the interface between the medical domain and the nursing domain. (...) But we are far from there, nurses are now taking the first steps to look at their data and what that can mean for healthcare practice.'* (Participant #10, Policy and Governance Expert)

## Theme 5: The role of the hospital in AI-development

Participants found the role of hospital in the development of AI pivotal, requiring a clear vision to bring AI technologies into clinical practice. In addition, this vision should be supported by the hospital taking the required practical steps in strategic planning and necessary infrastructure to make it happen.

First, participants found that the vision should fit with the current volume of patients treated in the hospital as they considered this is an important factor for developing adequate models. One participant said: '*You need a lot of patients to develop a good AI model. In our hospital we already have such volumes for certain diseases, so it is easier to develop a good model for those populations.'* (Participant #11, Policy and Governance Expert) Furthermore, participants found the vision also has to fit with the strategic goals of the hospital in the field of scientific research. As long as AI development is not part of this, awareness, facilitation and expertise building in the organisation is limited. In addition, collaboration with other hospitals with expertise on AI then remains occasional and project-based. One participant said: '*you need to start deploying AI in more than one part of the patient care pathway, then you can actually build knowledge expertise as well. (...) Otherwise it will remain point solutions.*' (Participant #2, healthcare professional)

Eventually, when the vision is formulated and aligned with the priorities, requirements can be met for the needed resource allocation to support this strategy. Determination of the technical prerequisites for successful AI implementation in the hospital was found as an important part of this strategy. One participant said: '*In other words, if I have to manually click on something with every patient that it has to be loaded, run out, results sent out and then reported (...) Those should actually be automatic. (...) And that's where we would worry about it now and try to enforce that this kind of automation we are going to find conditional.*' (Participant #2, healthcare professional) An example mentioned was the availability and openness of the data. As a result, choices should be made between structured and unstructured storage of data in patient records, with a structured approach facilitating the development of AI models. To document all this data, setting up a database to store this data is considered necessary. In addition, when the role of the hospital is also development of AI, investment in expensive and sophisticated computers are needed to develop AI models.

Another aspect discussed by several participants was the need of a governance on acceptable margins of error for AI models in patient care, given the lack of clarity and fear among professionals about responsibility and liability regarding data and models. A participant said: *'You also have a responsibility as a doctor or nurse with regard to the delivery of care. If AI is part of the delivery of care, I do need to understand how AI works for my decision-making so that I can have confidence in that as well. (...) If the algorithm does not work properly, then the question is who is responsible for that: the manufacturer or the doctor who provided the data.'* (Participant #10, Policy and Governance Expert) The lack of expertise in this area can make it difficult for professionals to bear this responsibility, especially when models are no longer transparent enough to understand. Therefore, the participant thought it seems to be essential to build knowledge on security, architecture and other technical aspects in order to set the right requirements for vendors and create a solid foundation for successful AI implementation in healthcare. Participants mentioned when such preconditions are not met, AI development would remain fragmented and not lead to real clinical benefits for patients. A participant said: *'There are so many initiatives. What are you going to commit to as a hospital, (...) what exactly are you going to do and why are you going to do it? (...) so far, I have actually missed that a little bit, so you see all kinds of very enthusiastic people who are very enthusiastic, but it remains in that phase'.* (Participant #2, healthcare professional)

A clear responsibility is outlined In the MDR legislation for the manufacturer of an AI as a medical device. Participants' opinions differed on whether or not the hospital should take this role. While some argued that hospitals possess valuable clinical expertise and firsthand knowledge of patient needs, positioning them as ideal collaborators or even leaders in AI development, others express concerns about potential conflicts of interest and diversion of resources away from core healthcare activities. Some participants felt that the goal of implementation of the model is for use in their own hospital level rather than implementation for other hospitals. This upscaling of implementation should be the role of commercial parties, as they have more expertise, funds to obtain certification for clinical use, and interests in commercialising the AI model. This contrast is well reflected in the following quotes: *'Yes, and who should then pick up that baton as being proof of that ambassador or that institute, who could that be? What kind of profile should that person or persons that group have, so are they colleagues healthcare background who also know how to translate the idea into a workable solution from the content or?'* (Participant #2, healthcare professional), while another said: *'You have to create certain conditions that say, 'OK, this AI is something that we can be responsible for and develop ourselves. And as soon as it goes beyond that, the hospital simply has to outsource it because you can no longer be responsible for it in terms of reliability, certification and so on.'* (Participant #11, Policy and Governance Expert).

### Theme 6: Opportunities and challenges from evolving laws and regulations

Participants highlight the current wave of legislative and regulatory developments which both provided opportunities and challenges for AI applications in healthcare. While these legislative and regulatory developments provide a promising framework for the safety and reliability of AI use, many also perceive barriers due to lack of clarity and unfamiliarity with the new regulations. This contrast was well reflected in the following quotes: *'so much is happening at the moment, at both national, European and global levels. We need to keep track of this, but also precisely map the integratability of the laws. How do they relate to each other?'* (Participant #13, Policy and Governance Expert) and *'I can hardly keep track of it all, let alone the physicians. Developments are moving very fast at the moment.'* (Participant #9, Policy and Governance Expert)

However, some even see opportunities in this dynamic, as it paves the way for safer use of AI technologies. In addition to these observations, participants argue for a balanced approach that allows room for innovation to flourish within the framework of findable, accessible, interoperable and reusable (FAIR) data principles. A clear vision and guidance from national governments on AI in healthcare is here for seen as necessary, independent of European Union regulations. One participant said: *'AI is currently still an innovation and it also needs space to develop without being completely boxed in. Of course, you must meet certain due care requirements but leave room for experimenting.'* (Participant #8, researcher).

Participants also emphasised that AI should only be included in clinical guidelines and regulations when there is sufficient evidence of its efficacy and benefits. AI applications should only be recommended after their added value to healthcare has been demonstrated, as they have the potential to provide better care to patients. They found that actual use of AI according the guidelines, should be reimbursed (e.g., financially) which could be a promoting factor to encourage AI in daily practice. In addition, it could also partially reimburse the costs incurred in AI development as well. One participant said: *'you should actually be rewarded if you use AI during your work to promote the adoption and implementation. Especially once these have been established in the guidelines.'* (Participant #1, healthcare professional)

## Discussion

This study highlights the complexity and diversity of challenges associated with AI implementation in clinical practice, identifying six themes that span value, application, technology, ethics, and responsibility. These themes provide a comprehensive framework for hospitals to enhance the potential value of AI for patients.

Key findings were the importance of aligning AI development with clinical needs rather than merely pushing new technologies. This alignment is crucial for successful clinical implementation. Therefore, it is essential to move beyond generating knowledge to creating clear pathways for implementing AI models. Timely involvement and effective communication with all stakeholders are also vital for successful AI implementation. In support, hospitals play a pivotal role in this process by providing a clear vision and strategy for AI development. This includes ensuring that the necessary infrastructure and expertise are in place. Additionally, staying aligned with evolving laws and regulations is important to ensure compliance and foster innovation. By focusing on these interconnected themes, hospitals can better navigate the complexities of AI implementation and enhance its value in clinical practice.

### Comparison with prior work

To our knowledge, this is the first study comprehensively explore the experiences of healthcare professionals, researchers, and Policy and Governance Experts regarding the practical implementation of AI in hospital settings. While previous studies have examined various challenges of integrating AI into healthcare, our research provides more in depth insights into the experiences by early adopters of AI technologies with the IFVAIH framework as guideline for the interviews.

Our findings have similarities and differences with previous studies that have explored the challenges of implementing AI in hospitals. One of the common barriers we identified is the complexity of successfully integrating AI into the workflow of healthcare providers. This is also acknowledged in a previous study, which highlights these challenges in applying AI to clinical workflows across Western and Middel Eastern countries [24–26]. However, a notable contrast between our findings and that study is the lack of fear of losing professional autonomy among the healthcare providers we interviewed in studies included in this review [27]. This may be due to the fact that our study focused on participants who are actively involved with AI implementation, resulting in more insights in the performance expectancy of AI implementation and less concerned about losing autonomy, but seeing it as an opportunity instead.

Furthermore, our findings are consistent with the challenges related to data quality and ethical policy issues, such as privacy, as discussed in another study, underscoring the universality of these issues when implementing AI in hospital settings [28]. For instance, data quality issues often arise from inconsistent data entry practices, lack of standardized protocols, and the integration of data from multiple sources (e.g. electronic health records and MRI scans in a separate viewer), which can lead to inaccuracies and hinder effective AI implementation. Additionally, related to ethical concerns and patient privacy are paramount, especially in regard to data accessibility for AI training. Nevertheless, these challenges are not unique to our study but are present in various implementation efforts of technological innovations, highlighting the need for robust data governance frameworks and ethical guidelines to ensure the responsible use of AI in healthcare.

Furthermore, the importance of policy and regulations and experience ethical and moral dilemmas with it were in line with previous studies from Europa, USA and Asia [29,30]. Although its importance was recognised, we found that

awareness among health professionals and scientists was limited, as was the involvement of relevant stakeholders. This limited awareness could be attributed to the rapid pace of advancements outstripping the development and dissemination of relevant laws and regulations and training programs. In addition, the complexity and specificity of regulations may deter thorough understanding and engagement, especially among healthcare professionals primarily focused on patient care.

Moreover, our results were in line with another Dutch study of Kim et al. [19], which highlights that healthcare professionals broader than radiology need transparency regarding the algorithms used in AI systems. This points to the essential role of understanding and explainability of the algorithms in building trust and acceptance of AI technologies among healthcare professionals which is also important in providing information to their patients [31].

Furthermore, our findings about the need for adequate interprofessional collaboration was in line with a previous review which also found teamwork as a relevant challenge for successful implementation of AI requiring strong communication, shared decision-making, coordinated actions, and progress evaluation eventually ensuring that different perspectives and expertise are integrated [32,33]. Also, emphasis on social influence and facilitating conditions further supports the need for strong interprofessional collaboration and organizational support [31]. However, achieving this level of collaboration can be challenging due to differences in professional languages, priorities, and workflows. Therefore, fostering a culture that promotes open communication, mutual respect, and continuous learning is essential for overcoming these barriers and ensuring the successful implementation of AI in healthcare settings. Here, a clear vision and strategy from the hospital may be required.

This role of the hospital to define a clear vision and strategy on AI development was also found in a previous study from Pakistan [25]). While this is supported by an earlier study [26], our findings highlight the need for hospitals to proactively establish the necessary infrastructure and expertise to support AI initiatives. By articulating a strategic approach to AI, hospitals can better align their resources and objectives with the technological advancements.

## Limitations

Strengths of the study include the depth, diversity, and quality of the data, which collectively provide a comprehensive and in depth view of current practice and allows for a nuanced understanding of the interplay between social and technical factors, highlighting the importance of considering both elements [34]. However, a few limitations that may influence the interpretation of the results. First, the context of this study was confined to one specific hospital. While this hospital is actively developing AI models and thus offers valuable insights, other hospitals with different infrastructures, cultures, and policy frameworks might encounter different challenges and experiences with AI implementation. Additionally, it may be of interest to include factors such as age and sex in a broader (quantitative) evaluation as these factors may be of influence in technology acceptance [31]. Nevertheless, it is important to note that generalizability is often not a primary aim of qualitative studies [35].

In addition, we attempted to include a variety of perspectives from different backgrounds, but mainly involved professionals working with AI on diagnostic images rather than AI based on unstructured data. Although this type of AI is most common in current practice, challenges experienced by stakeholders developing AI on unstructured chart data may be different. Additionally, more participants per subgroup could have further increased the depth and breadth of the findings. However, our current participants provided a comprehensive understanding of the various perspectives and experiences and data saturation was reached.

Another limitation is that this study used IFVAIH as the framework for the interview guide to understand the implementation of AI in clinical practice [36]. This model is specifically designed to provide practical guidance for AI implementation and while we used the IFVAIH funnel in data collection, the thematic analysis method allowed openness and adapt our approach to the themes that emerged during the coding process. This flexibility enabled us to uncover unexpected insights and patterns that might not have been captured if we had strictly adhered to the IFVAIH funnel. Nonetheless, other models are available, such as the Unified Theory of Acceptance and Use of Technology (UTUAT) and the Capability

Opportunity Motivation (COM-B) model. The exclusive use of the funnel may have resulted in missing insights that may be available through other theoretical frameworks such as the social influences [37].

## Implications

Based on the conducted study, the research team has gained several insights that could potentially be relevant for expediting AI implementation. Overall, it is crucial to focus on improvement of multiple domains rather than concentrating on a single aspect, as improvements can be achieved across all domains. Given this, we have identified the following implications which are covering all domains.

### Assessing the AI readiness of hospital

Drawing on themes 4 and 5, it becomes evident that maturity models may serve as valuable tools for evaluating the current state and charting a path forward [38]. By adopting a top-down approach to AI adoption, hospitals can gauge their maturity level and take targeted actions to enhance their AI capabilities. The Jaaksi & Jalavai model may be relevant here, as it involves assessing five key dimensions: data availability and quality, personnel expertise, existing solutions, and organizational readiness to adopt AI. Each dimension is divided into four levels, offering detailed insights into AI readiness and aligns with the domains Technology and Responsibility of the IFVAIH funnel [39].

### The hospital vision as a manufacturer of AI models

Drawing on themes 3, 4 and 5, predominantly domain Technology, Ethics and Responsibility, the study highlights the complex role of hospitals in AI development of AI. A key finding is the debate over whether hospitals should act like AI manufacturers. Some interviewees worry about conflicts of interest and resource allocation, while others advocate for hospitals' involvement due to their clinical insights and innovation potential. Ultimately, the level of hospitals' involvement in AI development may hinge on various factors, including their vision, institutional capacity, and expertise in the field of technology and ethics. Hospitals with abundant resources and specialized knowledge may be well-equipped to take on a manufacturer-like role. They can use their clinical insights to create customized AI solutions and secure funding for certification. Conversely, hospitals facing resource constraints or lacking in-house expertise may choose a more limited role, focusing on validating and implementing AI technologies developed by external parties.

### Standardizing the implementation process of AI tools

Given the uncertainty and unfamiliarity regarding the steps required to achieve AI implementation in practice based on themes 1 to 4, predominantly based on Application and Value, we recommend establishing a standard operating procedure (SOP) for evaluating and implementing AI solutions at the hospital level. This SOP would provide a structured approach to AI proposals and review, following standard steps to assess feasibility, ethical implications and clinical relevance of proposed AI applications. The use of the IFVAIH framework in this study can guide the development of such SOPs, although it is current impractical due to its heavy focus on technology. Here, the SOP should bridge the gap between grassroots initiatives and top-down management directives, fostering a collaborative and comprehensive approach to AI development. It is crucial, however, to ensure that these standardized procedures do not stifle innovation and experimentation. Creating space for innovative ideas and experimentation can lead to breakthroughs in AI implementation and drive continuous improvement in healthcare practices [40]. Also, distinguishing between research-oriented and implementation-oriented initiatives is important. These categories entail different SOPs. Research-oriented initiatives typically involve smaller-scale endeavours focused on knowledge generation, while implementation-oriented initiatives demand a broader spectrum of expertise and a nuanced understanding of the practical and implementation aspects of AI within clinical settings.

**Investing in mutual understanding between stakeholders**

Drawing on theme 3, an important aspect of successful AI implementation in hospitals it's crucial to foster long-term collaboration among physicians, technologists, researchers, and Policy and Governance Experts. Building enduring relationships is more beneficial than seeking short-term partnerships. To achieve this, integrate social learning into the SOP, such as using responsive evaluation to gather stakeholder perspectives, facilitate mutual learning, and assess AI impact [41]. Methods like action research or design thinking can help map changes and experiences. This includes discussing the interplay between social factors, such as stakeholder collaboration and organizational culture, and technical factors, such as IT infrastructure and data management [34]. Ultimately, establishing trust and expertise through these collaborations is key to successful implementation of AI initiatives.

## Conclusions

This study provides a unique broad, holistic perspective on the experiences of healthcare professionals, researchers and Policy and Governance Experts in optimizing implementation AI in hospitals, highlighting the complexity and diversity of challenges in implementing AI in clinical practice. It is essential to have a clear hospital vision and strategy that articulates the hospital's role in the development of AI. This will also enable the necessary resources to be allocated, allowing AI development to be considered from a variety of perspectives and expertise, increasing the potential for valuable AI in practice. Multidisciplinary collaboration, though currently limited, is vital and requires some form of facilitation. Balancing divergent perspectives and interests is crucial for ensuring the sustainability and successful implementation of AI.

## Supporting information

**S1 Appendix. Interview guide.**
(DOCX)

## Acknowledgments

The authors would like to thank all professionals participating in this study and all consortium partners of MEDIcal dAta analyTics cEnter (MEDIATE).

## Author contributions

**Conceptualization:** Jobbe PL Leenen, Paul Hiemstra, Martine M ten Hoeve, Joris D van Dijk, Brian Vendel, Guido Versteeg, Marike Hettinga.

**Data curation:** Jobbe PL Leenen, Martine M ten Hoeve.

**Formal analysis:** Jobbe PL Leenen, Paul Hiemstra, Martine M ten Hoeve, Anouk CJ Jansen, Gido A Hakvoort.

**Funding acquisition:** Gido A Hakvoort, Marike Hettinga.

**Investigation:** Jobbe PL Leenen, Paul Hiemstra, Martine M ten Hoeve.

**Methodology:** Jobbe PL Leenen.

**Software:** Jobbe PL Leenen.

**Supervision:** Gido A Hakvoort, Marike Hettinga.

**Validation:** Gido A Hakvoort.

**Writing – original draft:** Jobbe PL Leenen, Paul Hiemstra.

**Writing – review & editing:** Jobbe PL Leenen, Paul Hiemstra, Martine M ten Hoeve, Anouk CJ Jansen, Joris D van Dijk, Brian Vendel, Guido Versteeg, Gido A Hakvoort, Marike Hettinga.

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
