## [Decision Letter · Decision Letter 0]

2 Jan 2025

PDIG-D-24-00306Exploring the complex nature of implementation of artificial intelligence in clinical practice: an interview study with healthcare professionals, researchers and policy makersPLOS Digital Health Dear Dr. Leenen, Thank you for submitting your manuscript to PLOS Digital Health. After careful consideration, we feel that it has merit but does not fully meet PLOS Digital Health's publication criteria as it currently stands. Therefore, we invite you to submit a revised version of the manuscript that addresses the points raised during the review process. Please submit your revised manuscript within 30 days Feb 01 2025 11:59PM. If you will need more time than this to complete your revisions, please reply to this message or contact the journal office at digitalhealth@plos.org. Please include the following items when submitting your revised manuscript:* A rebuttal letter that responds to each point raised by the editor and reviewer(s). You should upload this letter as a separate file labeled 'Response to Reviewers '. This file does not need to include responses to any formatting updates and technical items listed in the 'Journal Requirements' section below.* A marked-up copy of your manuscript that highlights changes made to the original version. You should upload this as a separate file labeled 'Revised Manuscript with Track Changes '.* An unmarked version of your revised paper without tracked changes. You should upload this as a separate file labeled 'Manuscript '. If you would like to make changes to your financial disclosure, competing interests statement, or data availability statement, please make these updates within the submission form at the time of resubmission. Guidelines for resubmitting your figure files are available below the reviewer comments at the end of this letter. We look forward to receiving your revised manuscript. Kind regards, Aline Lutz de AraujoAcademic EditorPLOS Digital Health Leo Anthony CeliEditor-in-ChiefPLOS Digital Healthorcid.org/0000-0001-6712-6626  **Journal Requirements:**

1. In the online submission form, you indicated that Data will be available upon reasonable request at the corresponding author. 

a. In a public repository, 

b. Within the manuscript itself, or 

c. Uploaded as supplementary information.

 **Additional Editor Comments (if provided):** Dear Authors,

Thank you for submitting your manuscript to PLOS Digital Health. Three peer reviewers have assessed the quality and clarity of your work. They raised several points for clarification and highlighted the need to better connect the information presented across different sections.

The reviewers have requested further elaboration on the implications of your findings and a clearer integration of the discussion on AI implementation with your results.

We look forward to your revisions.

Kind regards,

Aline Lutz de Araujo

REVIEWER 1

Dear authors,

Thank you for your work on an interesting and highly relevant topic. In my opinion, there are some points that can be improved. These are detailed below.

Line 101: Please indicate the region in which this applies.

Line 148: I'm wondering whether policy-makers is the right term for this group.

Line 179: You should explicitly mention the qualitative data analysis approach you have chosen. Please justify the approach used.

Line 192: Make sure that the categories mentioned here are identical to the headings later on. In addition, it would be great to get an overview of all categories (“themes”) including subcategories (if there are any).

Line 201: You should briefly describe, what is meant by “Balancing between demand-pull and tech-push”. I assume that both of these terms were developed by you and were not directly mentioned by any of the interviewees.

Line 226: Is this quote really appropriate for the “tech-push” here? Please also check the quote in lines 239 and 245. It seems to me that you have a very broad understanding of demand, which should be specified.

Line 275: Please specify what "your project" means. Between the three sub-groups I would think it means quite different things. Overall, it would be interesting if similarities and differences between the subgroups were mentioned for all results.

Line 396: I would recommend that you summarize the main findings (sub-categories) before going into detail.

Line 424: In my opinion, it is not necessary for you to repeat the findings. It would be more interesting if you synthesized all the themes to find common patterns or relevant contextual factors. On this basis, you could improve the next section by not only comparing your findings with those of other studies, but also by providing explanations and theoretical assumptions.

Line 461: The fact that the participants are “pioneers of AI implementation” should be mentioned in the methods section as well as in the description of the study participants.

Line 466: Please explain in detail what is meant by “the importance of policy and regulations and experience ethical and moral dilemmas”.

Implication: I really appreciate that you want to give specific hints on how to implement AI. At the moment I am not able to connect this section with the results presented. The same problem exists with the IFVAIH model, here too I miss the connection to your results.

REVIEWER 2:

Overall, this is a topical and interesting study on the challenges of implementing AI in clinical practice. However, the manuscript would benefit from further proofreading. Some errors have been highlighted below, but this is not exhaustive.

Introduction.

• CE- marked? Please spell out acronym

• How do these five domains (Value, Application, ethics technology and responsibility ) map onto figure 1?

• Is figure 1 colour coded? (I.e does the red vs blue vs yellow have meaning)?

• Also, figure is repeatedly described as a “funnel” which suggests a pyramid structure, but then the figure is linear and implies the process is uni-directional.

Method

• It would be useful to get more information on the participant selection and sampling procedure, answering the following questions:

o Participants were contacted by email, but how were these emails obtained?

o Was the study advertised somewhere?

o Were participants known to the research group?

o Was there a screening survey used to identify eligible participants? Where their potential participants who were interested but not eligible?

• A flow diagram might be useful.

• In line with COREQ guidelines, could a small section on reflexivity be added- how did the interview moderators’ backgrounds influence the interview dynamic and analysis?

Results

• Age and gender play a large role in technology acceptance. It might be useful to add these columns to table. To minimise deductive disclosure, age categories rather than raw ages could be used. It would also be useful to reflect on how these factors influence the likelihood of being “early adopters” of AI.

• The sample has a wide range of professional diversity, which is great to see.

• Could an additional table showing the codes developed and which themes they informed be added? This would add greater transparency to the thematic analysis process.

• When giving participant quotes, could the participant professions be added alongside their ID, for increased readability and clarity?

• Line 253: should say “plan on how to”

• Line 379: should be a capital A, at the start of the sentence

Discussion

• Lines 515 to 518 are a repeat of lines 512 to 514.

• Under “comparisons with prior work” it would be useful if the geographical and temporal context was given to the studies cited, as developments and opinions of AI are happening at a rapid pace.

REVIEWER 3:

This is a very well written and concise manuscript that provides key findings for a current gap in the literature of AI innovation in clinical practice.

I have two broad comments for the discussion of the manuscript.

1. The authors identify six themes that are clearly defined and supported with the results. How do these challenges connect with the Innovation funnel presented in the introduction? It is valuable to bring the results into the framework. Perhaps linking each them/challenge to a stage for action in the funnel.

2. What are the strengths of the study? Future steps?

3. Authors could review a couple publications about the sociotechnical contexts. The context is mentioned as a limitation, how could this be explained/strengthened from a sociotechnical perspective.

4. Overall all sections are very well presented.

5. No suggestions of changes for the methodology. It is very well structured and corresponda to a high standard of qualitative research.**Reviewers' Comments:** Reviewer's Responses to Questions

**Comments to the Author**

1. Does this manuscript meet PLOS Digital Health’s publication criteria ? Is the manuscript technically sound, and do the data support the conclusions? The manuscript must describe methodologically and ethically rigorous research with conclusions that are appropriately drawn based on the data presented.

Reviewer #1: Yes

Reviewer #2: Yes

Reviewer #3: Yes

2. Has the statistical analysis been performed appropriately and rigorously?

Reviewer #1: N/A

Reviewer #2: N/A

Reviewer #3: N/A

3. Have the authors made all data underlying the findings in their manuscript fully available (please refer to the Data Availability Statement at the start of the manuscript PDF file)?

Reviewer #1: Yes

Reviewer #2: No

Reviewer #3: Yes

4. Is the manuscript presented in an intelligible fashion and written in standard English?

Reviewer #1: Yes

Reviewer #2: No

Reviewer #3: Yes

5. Review Comments to the Author

Reviewer #1: Dear authors,

Thank you for your work on an interesting and highly relevant topic. In my opinion, there are some points that can be improved. These are detailed below.

Line 101: Please indicate the region in which this applies.

Line 148: I'm wondering whether policy-makers is the right term for this group.

Line 179: You should explicitly mention the qualitative data analysis approach you have chosen. Please justify the approach used.

Line 192: Make sure that the categories mentioned here are identical to the headings later on. In addition, it would be great to get an overview of all categories (“themes”) including subcategories (if there are any).

Line 201: You should briefly describe, what is meant by “Balancing between demand-pull and tech-push”. I assume that both of these terms were developed by you and were not directly mentioned by any of the interviewees.

Line 226: Is this quote really appropriate for the “tech-push” here? Please also check the quote in lines 239 and 245. It seems to me that you have a very broad understanding of demand, which should be specified.

Line 275: Please specify what "your project" means. Between the three sub-groups I would think it means quite different things. Overall, it would be interesting if similarities and differences between the subgroups were mentioned for all results.

Line 396: I would recommend that you summarize the main findings (sub-categories) before going into detail.

Line 424: In my opinion, it is not necessary for you to repeat the findings. It would be more interesting if you synthesized all the themes to find common patterns or relevant contextual factors. On this basis, you could improve the next section by not only comparing your findings with those of other studies, but also by providing explanations and theoretical assumptions.

Line 461: The fact that the participants are “pioneers of AI implementation” should be mentioned in the methods section as well as in the description of the study participants.

Line 466: Please explain in detail what is meant by “the importance of policy and regulations and experience ethical and moral dilemmas”.

Implication: I really appreciate that you want to give specific hints on how to implement AI. At the moment I am not able to connect this section with the results presented. The same problem exists with the IFVAIH model, here too I miss the connection to your results.

Reviewer #2: Overall, this is a topical and interesting study on the challenges of implementing AI in clinical practice. However, the manuscript would benefit form further proofreading. Some errors have been highlighted below, but this is not exhaustive.

Introduction.

• CE- marked? Please spell out acronym

• How do these five domains (Value, Application, ethics technology and responsibility ) map onto figure 1?

• Is figure 1 colour coded? (I.e does the red vs blue vs yellow have meaning)?

• Also, figure is repeatedly described as a “funnel” which suggests a pyramid structure, but then the figure is linear and implies the process is uni-directional.

Method

• It would be useful to get more information on the participant selection and sampling procedure, answering the following questions:

o Participants were contacted by email, but how were these emails obtained?

o Was the study advertised somewhere?

o Where participants known to the research group?

o Was there a screening survey used to identify eligible participants? Where their potential participants who were interested but not eligible?

• A flow diagram might be useful.

• In line with COREQ guidelines, could a small section on reflexivity be added- how did the

• interview moderators’ backgrounds influence the interview dynamic and analysis?

Results

• Age and gender play a large role in technology acceptance. It might be useful to add these columns to table. To minimise deductive disclosure, age categories rather than raw ages could be used. It would also be useful to reflect on how these factors influence the likelihood of being “early adopters” of AI.

• The sample has a wide range of professional diversity, which is great to see.

• Could an additional table showing the codes developed and which themes they informed be added? This would add greater transparency to the thematic analysis process.

• When giving participant quotes, could the participant professions be added alongside their ID, for increased readability and clarity?

• Line 253: should say “plan on how to”

• Line 379: should be a capital A, at the start of the sentence

Discussion

• Lines 515 to 518 are a repeat of lines 512 to 514.

• Under “comparisons with prior work” it would be useful if the geographical and temporal context was given to the studies cited, as developments and opinions of AI are happening at a rapid pace.

Reviewer #3: This is a very well written and concise manuscript that provides key findings for a current gap in the literature of AI innovation in clinical practice.

I have two broad comments for the discussion of the manuscript.

1. The authors identify six themes that are clearly defined and supported with the results. How do these challenges connect withe Innovation funnel presented in the introduction? It is valuable to bring the results into the framework. Perhaps linking each them/challenge to a stage for action in the funnel.

2. What are the strengths of the study? Future steps?

3. Authors could review a couple publications about the sociotechnical contexts. The context is mentioned as a limitation, how could this be explained/strengthened from a sociotechnical perspective.

4. Overall all sections are very well presented.

5. No suggestions of changes for the methodology. It is very well structured and corresponda to a high standard of qualitative research.

6. PLOS authors have the option to publish the peer review history of their article (what does this mean? ). If published, this will include your full peer review and any attached files.

**Do you want your identity to be public for this peer review?** For information about this choice, including consent withdrawal, please see our Privacy Policy .

Reviewer #1: No

Reviewer #2: **Yes: ** Tosan Okpako

Reviewer #3: **Yes: ** Catalina Gonzalez Uribe

---

## [Decision Letter · Decision Letter 1]

3 Apr 2025

Exploring the complex nature of implementation of artificial intelligence in clinical practice: an interview study with healthcare professionals, researchers and Policy and Governance Experts

PDIG-D-24-00306R1

Dear Drs. Leenen,

We are pleased to inform you that your manuscript 'Exploring the complex nature of implementation of artificial intelligence in clinical practice: an interview study with healthcare professionals, researchers and Policy and Governance Experts' has been provisionally accepted for publication in PLOS Digital Health.

Best regards,

Dukyong Yoon

Section Editor

PLOS Digital Health

**Additional Editor Comments (if provided):**

**Reviewer Comments (if any, and for reference):**

Reviewer's Responses to Questions

**Comments to the Author**

1. If the authors have adequately addressed your comments raised in a previous round of review and you feel that this manuscript is now acceptable for publication, you may indicate that here to bypass the “Comments to the Author” section, enter your conflict of interest statement in the “Confidential to Editor” section, and submit your "Accept" recommendation.

Reviewer #2: All comments have been addressed

Reviewer #3: All comments have been addressed

2. Does this manuscript meet PLOS Digital Health’s publication criteria ? Is the manuscript technically sound, and do the data support the conclusions? The manuscript must describe methodologically and ethically rigorous research with conclusions that are appropriately drawn based on the data presented.

Reviewer #2: Yes

Reviewer #3: Yes

3. Has the statistical analysis been performed appropriately and rigorously?

Reviewer #2: N/A

Reviewer #3: N/A

4. Have the authors made all data underlying the findings in their manuscript fully available (please refer to the Data Availability Statement at the start of the manuscript PDF file)?

Reviewer #2: Yes

Reviewer #3: Yes

5. Is the manuscript presented in an intelligible fashion and written in standard English?

Reviewer #2: Yes

Reviewer #3: Yes

6. Review Comments to the Author

Reviewer #2: All comments have been addressed satisfactorily

Reviewer #3: All comments have been appropriately addressed. It is a remarkable manuscript that fills gaps in the literature.

7. PLOS authors have the option to publish the peer review history of their article (what does this mean? ). If published, this will include your full peer review and any attached files.

**Do you want your identity to be public for this peer review?** For information about this choice, including consent withdrawal, please see our Privacy Policy .

Reviewer #2: **Yes: ** Tosan Okpako

Reviewer #3: No
